# The Role of COA6 in the Mitochondrial Copper Delivery Pathway to Cytochrome *c* Oxidase

**DOI:** 10.3390/biom12010125

**Published:** 2022-01-13

**Authors:** Abhinav B. Swaminathan, Vishal M. Gohil

**Affiliations:** Department of Biochemistry and Biophysics, Texas A&M University, College Station, TX 77843, USA; abhinav22@tamu.edu

**Keywords:** mitochondria, copper, cytochrome *c* oxidase, COA6, COX2, Cu_A_ site, metallochaperone, disulfide reductase

## Abstract

Copper is essential for the stability and activity of cytochrome *c* oxidase (CcO), the terminal enzyme of the mitochondrial respiratory chain. Copper is bound to COX1 and COX2, two core subunits of CcO, forming the Cu_B_ and Cu_A_ sites, respectively. Biogenesis of these two copper sites of CcO occurs separately and requires a number of evolutionarily conserved proteins that form the mitochondrial copper delivery pathway. Pathogenic mutations in some of the proteins of the copper delivery pathway, such as SCO1, SCO2, and COA6, have been shown to cause fatal infantile human disorders, highlighting the biomedical significance of understanding copper delivery mechanisms to CcO. While two decades of studies have provided a clearer picture regarding the biochemical roles of SCO1 and SCO2 proteins, some discrepancy exists regarding the function of COA6, the new member of this pathway. Initial genetic and biochemical studies have linked COA6 with copper delivery to COX2 and follow-up structural and functional studies have shown that it is specifically required for the biogenesis of the Cu_A_ site by acting as a disulfide reductase of SCO and COX2 proteins. Its role as a copper metallochaperone has also been proposed. Here, we critically review the recent literature regarding the molecular function of COA6 in Cu_A_ biogenesis.

## 1. Cytochrome *c* Oxidase

Cytochrome *c* oxidase (CcO) is the terminal enzyme of the mitochondrial respiratory chain (MRC) that catalyzes the reduction of molecular oxygen to water using electrons from cytochrome *c* [1]. CcO is a multimeric protein complex present in the inner mitochondrial membrane and is composed of three core subunits encoded by the mitochondrial DNA and numerous (up to 11) accessory subunits encoded by the nuclear DNA (Figure 1a) [2,3]. Two of the core subunits of CcO, COX1 and COX2, contain metal cofactors that participate in electron transfer and catalysis (Figure 1b) [4,5]. The COX2 subunit contains two copper ions forming the binuclear Cu_A_ site and the COX1 subunit contains heme a, heme a_3,_ and a single copper ion, which together with heme a_3_ forms the Cu_B_/heme a_3_ site (Figure 1b). Together, these cofactors form the “molecular wire” that transfers electrons from reduced cytochrome *c* to molecular oxygen. The process of electron transport is coupled to proton pumping from the matrix to the intermembrane space (IMS), an activity carried out by other complexes (I and III) of the MRC as well. The proton pumping activity of these MRC complexes generates a proton gradient across the inner mitochondrial membrane, which drives mitochondrial energy production in the form of adenosine triphosphate (ATP).

The copper requirement of CcO necessitates its delivery to the mitochondria [6]. With copper being a highly reactive metal, with its ability to generate deleterious reactive oxygen species and displace other metals from their native coordination sites [6,7,8,9,10], cells employ special strategies to transport and insert copper into target proteins. This involves small copper-binding molecules as well as proteins termed metallochaperones, which bind copper and insert it into its target proteins via specific protein–protein interactions [11].

## 2. The Mitochondrial Copper Delivery Pathway

Cytosolic copper is delivered to the mitochondrial matrix in a ligand-bound form, called copper-ligand (CuL), though the molecular identity of this ligand remains elusive [12]. Although copper insertion into the CcO subunits occurs in the mitochondrial IMS (Figure 2), cytosolic copper is first transported into the mitochondrial matrix via the mitochondrial phosphate carrier protein Pic2 in yeast and its mammalian homolog SLC25A3 [13,14]. Pic2-deleted yeast does not exhibit a robust copper-deficiency phenotype [13,15], suggesting alternate mechanism(s) of copper transport into the mitochondria. Indeed, it was later shown that mitochondrial iron transporter, Mrs3, could also transport copper to the mitochondrial matrix [16]. The mitochondrial matrix copper pool is the main source of copper ions that are inserted into the CcO subunits [12]. Thus, mobilization of copper from the mitochondrial matrix to the IMS for its delivery to CcO copper sites requires its efflux back to the IMS through a yet unidentified transporter [17]. Once in the IMS, a number of evolutionarily conserved proteins participate in its insertion into the COX1 and COX2 subunits of CcO [18] (Figure 2). Genetic and biochemical studies have shown that IMS-localized COX17 is the common source of copper for the COX1- and COX2-specific metallochaperones [18]. The next steps in copper delivery to CcO are a modular process, which involves the biogenesis of Cu_A_ and Cu_B_ as two separate modules on COX2 and COX1, respectively (Figure 2) [19,20]. Before copper insertion into the Cu_B_ site, heme cofactors are added on to the COX1 subunit, which is inserted into the inner mitochondrial membrane [18]. Similarly, copper insertion into the Cu_A_ site requires the membrane insertion of COX2 and the subsequent translocation of its copper-binding domain to the mitochondrial IMS [18]. This topology of COX1 and COX2 copper-binding domains necessitates the IMS localization of the CcO assembly factors involved in copper delivery (Figure 2). Within the IMS, SCO1, SCO2, and COA6 are involved in copper delivery to the COX2 module whereas COX11 and COX19 are involved in copper delivery to the COX1 module (Figure 2) [18,21,22,23,24,25,26,27,28,29,30]. The localization of the copper delivery proteins in the oxidizing environment of the IMS is reminiscent of prokaryotes, where copper delivery into the CcO subunits occurs in the periplasmic space.

The precise molecular functions of metallochaperones COX17, SCO1, and COX11, have been determined and they have been shown to transfer copper to CcO subunits in a bucket-brigade fashion [18,25,26,27,29,30]. Specifically, COX17 receives copper in the IMS via an unknown mechanism and transfers it to COX11 and SCO1, which then metallates copper sites on COX1 and COX2, respectively (Figure 2) [25,26,27,29,30]. In addition to these metallochaperones, SCO2, COA6, and COX19 have also been shown to be a part of the copper delivery pathway in the IMS [21,23,24,28,29]. SCO2, which is a paralog of SCO1, also receives copper from COX17 and acts as a copper-dependent disulfide reductase of COX2, a reaction necessary for its metallation by SCO1 [29]. COX19 has been shown to play a role in maintaining the redox state of COX11 and facilitating copper delivery to the COX1 subunit [28]. COA6 facilitates the assembly of the Cu_A_ site of COX2 [21,22,23,24], and its precise biochemical function in this process is the focus of this review.

## 3. History of Discovery of COA6

COA6, originally named C1ORF31 in humans and Ymr244C-A in yeast, was first identified as a putative CcO assembly factor based on an iterative orthology prediction [31]. Subsequently, it was found in a proteomic survey of mitochondrial IMS proteins in the yeast *Saccharomyces cerevisiae* and was renamed Coa6 (Cytochrome Oxidase Assembly factor 6) because of its requirement for CcO assembly [32]. Although these studies were the first to link Coa6 to CcO biogenesis, its precise role in the process remained unclear. Based on its sub-mitochondrial localization, human genetics, and protein sequence, we predicted the role of Coa6 in the copper delivery pathway to CcO [21]. Three key findings from our group confirmed this prediction. First, copper supplementation in the yeast growth media rescued respiratory growth and CcO assembly of *coa6∆* cells, providing the first link between Coa6 and the mitochondrial copper delivery pathway to CcO [21]. Second, a comprehensive genetic interaction study uncovered a synthetic lethal interaction between yeast *coa6∆* and *sco2∆* mutants, a finding that placed Coa6 in the copper delivery to the Cu_A_ site [23]. Third, co-immunoprecipitation of Coa6 identified specific interaction of Coa6 with proteins involved in the Cu_A_ site biogenesis, including Sco1, Sco2, and Cox2 [23]. We further showed that heterologous expression of human COA6 was able to rescue the respiratory growth of yeast *coa6∆* indicating that the function of Coa6 is evolutionarily conserved [23]. Indeed, a subsequent study with *COA6* patient fibroblasts recapitulated copper-mediated rescue of COX2 levels [33]. Consistent with our studies with the yeast model system, two other groups independently demonstrated the role of COA6 in COX2 biogenesis in mammalian cells through protein–protein interaction studies [22,34]. Specifically, both groups found that human COA6 interacts with the newly synthesized COX2 [22,34], though there was some discrepancy between these two studies regarding the interaction of COA6 with SCO proteins, with one group reporting COA6:SCO1 interaction [22] and the other group reporting COA6:SCO2 interaction in vivo [34]. Our recent in vitro studies with purified human proteins have shown that COA6 can interact with both SCO1 and SCO2; however, when all three proteins are present in the reaction mixture, COA6 preferentially interacts with SCO1 [24]. These studies together, unequivocally place COA6 in the final steps of Cu_A_ site biogenesis.

## 4. Structure of COA6

The structure of the human recombinant COA6 protein has been determined by both X-ray crystallography and nuclear magnetic resonance (NMR) spectroscopy. Both the structures show COA6 as a helical protein with coiled-coil-helix-coiled-coil-helix (CHCH) fold domain [24,35,36]. COA6 contains three α-helices where the first two helices—helix 1 (H1) and helix 2 (H2)—form a CHCH domain stabilized by two disulfide bonds between the cysteine pairs of its CX_9_C-CX_10_C motif (Figure 3a,b) [24,36]. Despite these similarities between the solution and crystal structures, a number of differences exist, as described in detail in a recent review [35]. Briefly, the length of the H1 and H2 helices in the crystal structure is significantly longer than those in the solution structure (Figure 3a,b). In the solution structure, helix 3 (H3) is broken into a shorter fourth helix and the orientation of H3 is also different between these two structures (Figure 3a,b) [24,36]. In the crystal structure, the H3 of one monomer of COA6 interacts with the other, thereby stabilizing the dimeric state whereas in the solution structure this helix is present at the opposite face of the CHCH domain (Figure 3a,b) [24,36]. Notably, the structure of COA6 predicted through AlphaFold software is more similar to the crystal structure of COA6. We speculate that the indicated differences between the crystal structure and the solution structure could be due to the dynamic nature of the protein and its concentration dependent oligomerization involving H3 [24]. In support of this idea, NMR-based analysis of the interaction of COA6 with SCO1 reveals that the residues of H3 in COA6 are also involved in its interaction with SCO1 [24]. Therefore, it appears that this third helix is dynamic and plays a crucial role not only in the dimerization of COA6 but also in its interaction with its target proteins. Indeed, deletion of H3 abrogated COA6 function in vivo [24].

The proteins containing the CHCH domain have previously been shown to perform diverse functions in the cell [27,37,38]. For example, two well-characterized CHCH-domain containing proteins, COX17 and MIA40, function as metallochaperone and thiol-disulfide oxidoreductase, respectively. In the case of COX17, two cysteines at the N-terminal region outside the CHCH domain bind copper (Figure 3c), which is necessary for its role as a copper metallochaperone [27,39]. MIA40 is part of the protein import machinery of the mitochondrial IMS that functions as a thiol-disulfide oxidoreductase. It contains redox-active cysteines in its CPC motif present at the N-terminal region of the protein outside its CHCH domain (Figure 3d) [38]. Based on the known functions of these CHCH-domain containing proteins, COA6 can be predicted to act as a metallochaperone or a disulfide reductase.

## 5. Is COA6 a Copper Metallochaperone?

Metallochaperones are defined as proteins that bind metal ions and deliver them directly to target protein(s) via specific protein–protein interactions [11]. Based on this definition, if COA6 is a copper metallochaperone then it is expected to bind copper, physically interact with its target protein(s), and insert copper into the metal-binding sites on the target protein(s). Below, we critically evaluate these criteria for the potential metallochaperone role of COA6.

There is discrepancy in the literature regarding the copper-binding ability of COA6, its stoichiometry of binding, and the physiological relevance of copper binding. One study showed that purified GST-Coa6 fusion protein could bind copper at a mass ratio of about 8.5 ng copper per µg of protein [34]. This would equate to copper:Coa6 molar stoichiometry of about 5:1. In the same study, GST-Sco1 and GST-Sco2, proteins known to bind copper at a 1:1 molar ratio, were used as controls and were shown to have copper: protein molar stoichiometries of about 6:1 and 5:1, respectively. Clearly, the experimental conditions used in this study were skewed towards excess copper binding. Another group performed a competitive in vitro copper-binding assay by incubating purified human COA6 with Cu(I)-bathocuproine disulfonic acid (BCS) complex and determined that COA6 binds Cu with a very high affinity (K_d_ of ~10^−17^ M) [22]. They further showed that COA6 binds copper in a 1:1 stoichiometry and mutagenesis of the first and fourth cysteines of COA6 completely eliminates copper binding [36]. In contrast to these studies, we determined that purified recombinant human COA6 did not contain any bound copper [24]. Nevertheless, the reconstitution of purified COA6 with copper under reducing conditions, as described in the previous studies [22,36], did result in copper binding [24]. However, the ability of COA6 to bind copper in vitro could be an experimental artifact since copper being thiophilic, is expected to bind to the cysteine residues of COA6 that are forcibly reduced. In fact, a similar proposal was made for Cmc1, another twin CX_9_C motif-containing mitochondrial IMS protein, which was shown to bind copper upon in vitro reconstitution [40]. Later, it was found to have a copper-independent role in the stabilization of the COX1-intermediate [37]. Thus, the Cmc1 work serves as a cautionary example of using an in vitro copper reconstitution assay as a metric for linking the function of twin CX_9_C motif-containing proteins to copper metabolism. Moreover, the concentration of free copper ions inside the cell is virtually zero since all copper is either protein bound or is sequestered by non-proteinaceous anionic ligands [12,41]. Therefore, the in vitro reconstitution studies do not mimic in vivo conditions. While demonstrating in vivo binding of copper to protein is challenging, our work using online tandem liquid chromatography-inductively coupled plasma-mass spectrometry (LC-ICP-MS)-based experiments suggested that Coa6 is not bound to copper in vivo [24].

Several additional observations call into question the metallochaperone role of COA6. First, COA6 lacks the canonical copper-binding residues present in other CHCH-domain containing copper metallochaperones such as COX17 (Figure 3a–c). Although it was proposed that copper binds through the first and fourth cysteines of the CX_9_C-CX_10_C motif of COA6 [36], the cysteines of the CX_9_C-CX_9-10_C motif have not been shown to bind copper in other well-characterized proteins such as COX17, COX6B, and MIA40. Notably, none of the structures reported for COA6 are in its copper-bound form [24,36]. Second, if COA6 acts as a copper metallochaperone in copper delivery to COX2, its deletion should phenocopy *cox17∆* or *sco1∆* yeast mutants because deletion mutants of all metallochaperones involved in copper delivery to CcO display strong respiratory deficient phenotypes. In contrast, *coa6∆* exhibits much milder respiratory phenotypes including partially reduced respiratory growth, respiration, and CcO abundance [21,23]. Third, while protein–protein interaction studies in both yeast and mammalian cells have demonstrated interaction of COA6 with SCO1, SCO2, and COX2, currently no studies have shown the direct transfer of copper from COA6 to its interacting partners [22,23,34]. Without this direct evidence, it is too premature to call COA6 a copper metallochaperone. Taken together, these studies argue against a metallochaperone role for COA6 in vivo.

## 6. Is COA6 a Disulfide Reductase?

A prerequisite for copper binding by the cysteine ligands of the cuproproteins is that they must be in a reduced (sulfhydryl) state. The IMS of mitochondria has a reduction potential of −255 mV [42] and the reduction potential of copper-binding cysteines of SCO1 and COX2 were shown to be about −280 mV and −290 mV, respectively [29,43]. Now, assuming thermodynamic equilibrium, we would expect that these cysteines would be prone to oxidation in the IMS. This would pose a problem in copper transfer to the Cu_A_ site and would require the action of disulfide reductase(s) that can reduce these copper-binding cysteines. Bacteria face a similar problem because the copper-binding cysteine residues in the bacterial Cu_A_ site lie on the periplasmic side of the cytosolic membrane, which is also an oxidizing environment. These copper-coordinating cysteines are prone to form disulfide bonds, which must be reduced prior to copper insertion. The periplasmic thioredoxin, TlpA, was shown to perform this role by reducing the copper-binding cysteines of both SCOI and COXB, the bacterial homologs of SCO1 and COX2, respectively [44]. As described below, recent studies show that COA6 performs an analogous role in the eukaryotic mitochondria by reducing the disulfides of both SCO1 and COX2 [24].

A number of observations hinted at the potential redox role of COA6. First, the respiratory growth phenotype of yeast *coa6∆* mutant was suppressed under reducing conditions [24]. Second, yeast *COA6* exhibited a strong synthetic lethal interaction with *SCO2*, which has already been shown to act as a disulfide reductase in vitro and in vivo [23,45]. Third, CHCH domain-containing proteins have been proposed to function as redox proteins [46]. In line with these observations, it was shown that COA6 can reduce the disulfides of SCO1, SCO2, and COX2, in vitro [24]. Importantly, this reaction is thermodynamically favored since the reduction potential values of COA6, variously reported as −330 mV [24] or −349 mV [36], is lower than that of SCO1 (−280 mV), SCO2 (<−300 mV), and COX2 (−290 mV) [24]. Consistent with this in vitro data, it was found that COA6 is required to maintain the cysteines of SCO1 and SCO2 in the reduced form in vivo [24,47]. Thus, there is strong experimental evidence in support of the disulfide reductase activity of COA6.

Despite this evidence, the molecular basis of disulfide reductase activity of COA6 is not known and some observations are not consistent with this role. For example, although COA6 has a CHCH domain similar to that seen in another thiol-disulfide oxidoreductase, MIA40, the redox-active CPC motif of MIA40 is absent in COA6. The cysteine residues of CX_9_C-CX_10_C motif of COA6 are likely candidates for the disulfide reductase activity. However, there is no evidence that the other CX_9_C-CX_9-10_C motif-containing proteins, such as MIA40 or COX17, exhibit redox activity using the cysteines of the CX_9_C-CX_9-10_C motif. Further, since COA6 has been shown to be a MIA40 substrate [32], at least one pair of its cysteines will be in the oxidized form upon its import into the mitochondrial IMS, making them incapable of redox cycling. Contradictory evidences exist regarding the identity of these potential redox-active cysteine residues. NMR studies on COA6 showed that the second and third cysteine pairs are capable of redox cycling [24]. However, a recent study showed that the first and fourth cysteine pairs of COA6 are redox active with a reduction potential of −349 mV [36]. In addition to this discrepancy, the NMR chemical shift perturbation data on the interaction between SCO1 and COA6 showed that only helix 3 of COA6 is involved in the intermolecular interaction with SCO1. None of the cysteines of COA6 or the copper-binding cysteines of SCO1 were shown to be involved in this interaction. Importantly, the source of electrons used by COA6 for its disulfide reductase activity is unknown. Taken together, the molecular mechanism by which COA6 influences the redox state of SCO1 and COX2 still remains unclear.

## 7. Concluding Remarks

The delivery and insertion of copper cofactors in the mitochondrial CcO requires a number of “assembly factors”. Elucidating the biochemical function of these CcO assembly factors has remained a major bottleneck in our understanding of the biogenesis of CcO, with the role of COA6 in Cu_A_ site formation being a case in point. A recent review article debated the potential role of COA6 as a metallochaperone or a thiol-disulfide oxidoreductase (disulfide reductase) in the copper delivery process [35]. Upon critical analysis of literature, we find that the current experimental data favors the disulfide reductase activity of COA6 and leads to a model where COA6 acts on its client proteins—COX2 and SCO1—by reducing the disulfide bond between their copper-coordinating cysteines to allow for copper binding (Figure 4). Future studies are required to determine the COA6 residues that participate in the disulfide reductase activity, the mechanism of its substrate specificity, and the structural basis of its activity. Addressing these issues could establish COA6 as a paradigm for understanding the biochemical roles of other CX_9_C-CX_9-10_C motif containing proteins in CcO biogenesis.

## Figures and Tables

**Figure 1 biomolecules-12-00125-f001:**
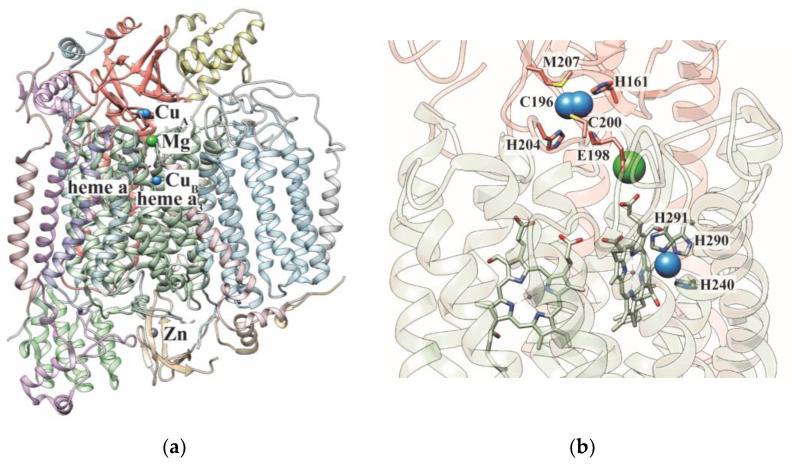
Structure of human cytochrome *c* oxidase. (**a**) The structure of the 14-subunit human cytochrome *c* oxidase is shown (PDB: 5Z62). The Cu_A_ site is present on COX2 subunit (salmon) and heme a, heme a_3_/Cu_B_ sites are present on COX1 subunit (green). In addition to this, the enzyme also binds other metals such as Mg and Zn. (**b**) The copper-containing subunits are shown. The coordination ligands of the copper centers are depicted as sticks. The Cu, Mg, and Zn metals are shown as blue, green, and grey spheres, respectively.

**Figure 2 biomolecules-12-00125-f002:**
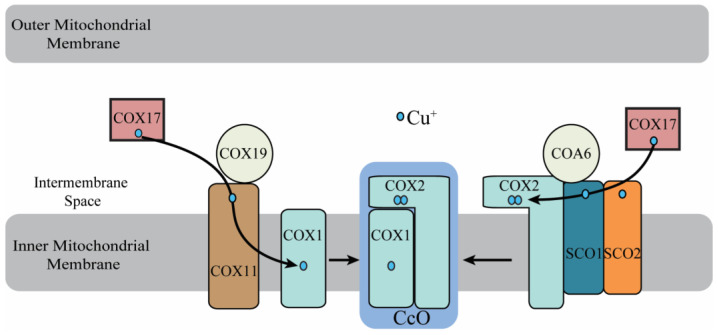
A schematic diagram of the mitochondrial copper delivery pathway to cytochrome *c* oxidase. The copper insertion into CcO is a modular process where COX1 and COX2 metallation occurs separately using a dedicated set of proteins. COX17 is the primary donor of copper to both COX1 and COX2, directly delivering copper to COX11 and SCO proteins. SCO1 is the metallochaperone that inserts copper into COX2, with SCO2 and COA6 facilitating the copper transfer. COX11 is the metallochaperone that inserts copper into the COX1 subunit and requires COX19 for its functionality.

**Figure 3 biomolecules-12-00125-f003:**
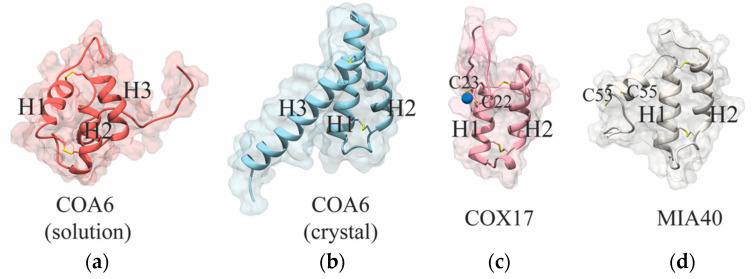
Comparison of the structure of COA6 with other human CHCH domain-containing proteins. The solution structure of COA6 (PDB 6NL3) (**a**) and a monomer from the crystal structure of dimeric COA6 (PDB 6PCE chain B) are shown (**b**). Solution structure of COX17 (PDB 2RNB) (**c**). Solution structure of human MIA40 (PDB 2K3J) (**d**). The cysteines that form the disulfide bonds in the CX_9_C-CX_9-10_C motif have been shown in sticks. The copper binding cysteines of COX17 (**c**) and the redox-active cysteines of MIA40 (**d**) have also been highlighted in sticks. Copper is shown as a blue sphere.

**Figure 4 biomolecules-12-00125-f004:**
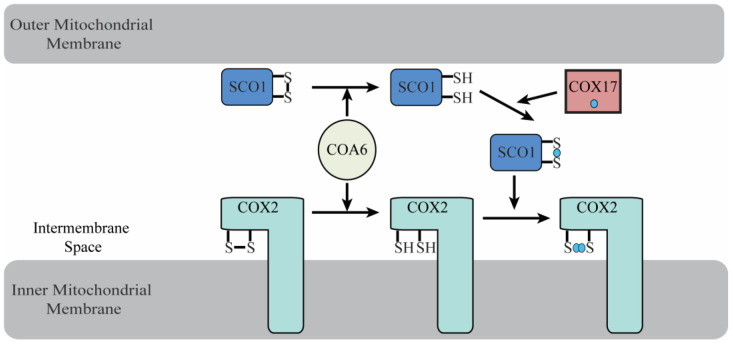
A schematic diagram of the biochemical role of COA6 in the metallation of COX2. The metallation of SCO1 and COX2 requires that their copper-binding cysteines are in the reduced form. COA6 acts as a disulfide reductase in this process, reducing the cysteines of both oxidized SCO1 and COX2, enabling their copper metallation by Cu-COX17 and Cu-SCO1, respectively.

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
