# Peer review of "The Role of COA6 in the Mitochondrial Copper Delivery Pathway to Cytochrome c Oxidase"

_biomolecules, 2022, doi:10.3390/biom12010125_

Round 1

Reviewer 1 Report

This review describes the role of COA6 protein in the mitochondrial copper delivery pathway to cytochrome c oxidase. It is well-written and clearly presents the possible role of COA6 as copper chaperone or disulfide reductase, taking well into account all literature data. I fully agree with the authors that the in vivo and in vitro available data argue against a metallochaperone role for COA6 in vivo. I also appreciate that the authors present a critical view on the still undefined role of the COA6 residues that participate in the disulfide reductase activity. On this regard, I have only a minor comment that should be presented in the text: the authors should mention that the still open question on COA6 function might be positively addressed by defining whether COA6 is a substrate of Mia40. Mia40 might indeed introduce both or just one disulfide in the CX9C-CX10C motif. Thus, this information can help to define whether one Cys motif between the two present, i.e. that one being not oxidized by Mia40, can work in the disulfide reductase function.

Another minor point:

the reduction potential of COA6 reported in the text is referred to both disulfides or to just one of them? This should be specified if possible on the basis of the published literature data.

Reviewer 2 Report

The manuscript by Swaminathan & Gohil is a very well-written review by the true experts in the field covering current state-of-the-art in copper delivery and incorporation into one of the key components of the mitochondrial respiratory chain, cytochrome c oxidase (CcO). Although intensively studied over the last years, the exact copper delivery system and associated steps of CcO assembly remain still incompletely understood. Particularly, the role of COA6 in this process is under intensive debate. This review summarises the current knowledge in the field and critically discusses the plausible function of the controversial COA6 in CcO biogenesis. Together with another recently published review paper (PMID:32977416) it provides a complete and comprehensive discussion on the role of this important, disease-associated factor. Manuscript by Swaminathan & Gohil will be certainly a precious piece of knowledge for both specialists in mitochondrial respiratory chain function and metalloprotein biochemistry. Therefore, I strongly support this review for publication in Biomolecules and have only minor suggestions for consideration by the authors.

  1. Authors discuss extensively how copper ions are delivered to the protein targets at the CcO. Yet, very little is said on how copper delivery is entangled in the global CcO assembly process (e.g., at which stage of CcO assembly is copper incorporated and whether there is any quality control mechanism/checkpoint involved, etc.). Could it also support some of the claims made by authors on the role of COA6? The brief comments on this matter could be of great value for readers with less expertise in the topic.
  2. Can authors shortly speculate why copper must be delivered to IMS via back-route from the mitochondrial matrix to support CcO biogenesis? How much has the elaborate copper delivery system evolved to compensate for copper instability issues imposed by the IMS microenvironment? This is an intriguing and barely discussed phenomenon.
  3. Figures are well-designed and very informative. Yet, Figure 1 is a bit difficult to read. Could authors consider the bigger and more visible font for panel 1(a). Panel 1(b) could also benefit from zooming directly into the copper-binding site. There might also be a need to explain the color coding for the sphere representation of different metal ions.
  4. The authors discuss a compelling discrepancy between the crystal and solute COA6 structure broadly. I wondered whether the authors also tried to predict the COA6 structure using computational algorithms as the AlphaFold. To which model would it suit better?
